# The Sweetpotato Voltage-Gated K^+^ Channel β Subunit, KIbB1, Positively Regulates Low-K^+^ and High-Salinity Tolerance by Maintaining Ion Homeostasis

**DOI:** 10.3390/genes13061100

**Published:** 2022-06-20

**Authors:** Hong Zhu, Xue Yang, Qiyan Li, Jiayu Guo, Tao Ma, Shuyan Liu, Shunyu Lin, Yuanyuan Zhou, Chunmei Zhao, Jingshan Wang, Jiongming Sui

**Affiliations:** 1College of Agronomy, Qingdao Agricultural University, Qingdao 266109, China; zhuhong198812@126.com (H.Z.); yangxue12072022@163.com (X.Y.); lqy183292@163.com (Q.L.); 17852845051@163.com (J.G.); mataoqau@163.com (T.M.); rr50053180@163.com (S.L.); L17852843951@163.com (S.L.); meiwei2002@163.com (C.Z.); jswang319@163.com (J.W.); 2Laboratory of Microbiology, Institute of Biology, Hebei Academy of Sciences, Shijiazhuang 050081, China; 3Crop Research Institute, Shandong Academy of Agricultural Sciences/Scientific Observing and Experimental Station of Tuber and Root Crops in Huang-Huai-Hai Region, Ministry of Agriculture and Rural Affairs, Jinan 250100, China; zhou_yy_2020@163.com

**Keywords:** sweetpotato, voltage-gated K^+^ channel β subunit, KIbB1, low K^+^, high salinity, ion homeostasis

## Abstract

Voltage-gated K^+^ channel β subunits act as a structural component of K_in_ channels in different species. The β subunits are not essential to the channel activity but confer different properties through binding the T1 domain or the C-terminal of α subunits. Here, we studied the physiological function of a novel gene, *KIbB1*, encoding a voltage-gated K^+^ channel β subunit in sweetpotato. The transcriptional level of this gene was significantly higher in the low-K^+^-tolerant line than that in the low-K^+^-sensitive line under K^+^ deficiency conditions. In *Arabidopsis*, *KIbB1* positively regulated low-K^+^ tolerance through regulating K^+^ uptake and translocation. Under high-salinity stress, the growth conditions of transgenic lines were obviously better than wild typr (WT). Enzymatic and non-enzymatic reactive oxygen species (ROS) scavenging were activated in transgenic plants. Accordingly, the malondialdehyde (MDA) content and the accumulation of ROS such as H_2_O_2_ and O^2−^ were lower in transgenic lines under salt stress. It was also found that the overexpression of *KIbB1* enhanced K^+^ uptake, but the translocation from root to shoot was not affected under salt stress. This demonstrates that *KIbB1* acted as a positive regulator in high-salinity stress resistance through regulating Na^+^ and K^+^ uptake to maintain K^+^/Na^+^ homeostasis. These results collectively suggest that the mechanisms of *KIbB1* in regulating K^+^ were somewhat different between low-K^+^ and high-salinity conditions.

## 1. Introduction

The salinity of agricultural lands is one of the most serious abiotic stresses limiting plant growth and crop yield [1]. Due to their sessile growth habits, plants are inevitably exposed to environmental stimuli during their life cycles [2]. To cope with high-salinity stress, plants can assume a different geographical distribution over a long period of time [3]. A high NaCl concentration induces a series of physiological and biochemical changes in plants [4,5]. The destruction of ROS and ion homeostasis leads to a range of physiological, metabolic and genomic impairments [6]. It has been demonstrated that the accumulation of Na^+^ rather than Cl^−^ causes toxic effects in the cytoplasm in most situations [7]. Potassium (K^+^) is the second most abundant inorganic cation in plant tissues, accounting for 2–10% of the dry weight [8]. K^+^ plays important roles in many physiological and metabolic processes such as enzyme activation, stomatal movement, protein synthesis, osmoregulation, and responses to biotic and abiotic stress [9]. The deficiency of K^+^ impairs photosynthesis and sugar accumulation, which detrimentally affects plant quality and yield [10]. The metabolic toxicity of Na^+^ is due to its ability to compete with K^+^ for binding sites and thus destroy the well-balanced cellular function [6]. Meanwhile, the high level of Na^+^ can inhibit K^+^ acquisition and cause osmotic imbalance [11,12]. Therefore, maintaining a high level of the K^+^/Na^+^ ratio is one of the most effective ways to alleviate the toxicity caused by salt stress [4,13]. Since K^+^ is not metabolized in plant cells, its regulation is mainly dependent on the uptake from the environment and translocation in plants [14]. In plants, K^+^ uptake is mainly mediated by K^+^ channels and K^+^ transporters [15,16].

K^+^ channels consist of two main categories, voltage-gated K^+^ channels and voltage-independent K^+^ channels [17,18]. These two kinds of K^+^ channels have different mechanisms for mediating K^+^ flux [19]. Voltage-gated K^+^ channels are important transmembrane regulators selectively catalyzing the transport of K^+^ across the membrane [20]. The first voltage-gated K^+^ channel gene was cloned from *Drosophila* [21]. Since then, several members have been discovered and evaluated in plants. A total of nine different voltage-gated K^+^ channels were identified in the genome of *Arabidopsis thaliana* [22,23]. According to the structure and function, these members are grouped into four clades: outward-rectifying K^+^ (K_out_) channels, including SKOR and GORK; inward-rectifying K^+^ (K_in_) channels, including KAT1, KAT2, AKT1, SPIK and AKT5; weak-rectifying K^+^ (K_weak_) channels including AKT2; and silent-rectifying K^+^ (K_silent_) channels including KC1 [24]. The K_silent_ channels only modulate channel properties and do not form functional channels [25]. The K_in_ channels are divided into two subclades: the K_in_-a clade (KAT1-like) and K_in_-b clade (AKT1-like) [26,27]. The functions of K_in_ channels have been widely analyzed in many kinds of plant species. In *Arabidopsis*, the expression level of *AtAKT1* was induced by K^+^ deficiency, and the *AtAKT1* mutation significantly reduced K^+^ acquisition [28,29]. These results indicate that AtAKT1 played an important role in mediating K^+^ uptake into roots. In rice, a mutation of *OsAKT1* resulted in sensitivity to low-K^+^ stress, and *OsAKT1* was also related to salt stress tolerance [30]. In barley, the induced expression of *HvAKT1* might lead to K^+^/Na^+^ homeostasis under high-salinity stress [31]. The overexpression of *GmAKT1* enhanced K^+^ uptake and K^+^/Na^+^ balance to regulate tolerance to low-K^+^ and salt stress in soybean [32]. PutAKT1, isolated from *Puccinelli tenuiflora*, could increase K^+^ content to regulate salt stress tolerance in transgenic *Arabidopsis* [33]. Previous studies reported that the voltage-gated K^+^ channels contained four α subunits forming an ion-conducting pore [34]. Subsequently, a second type, the β subunit, acting as a structural component of K_in_ channels, was found in different species [35,36,37,38]. The β subunits are not essential to the channel activity but confer different properties [38]. The β subunits have been proved to change the inactivation rate and promote the maturation and stabilization of their target channels through binding the T1 domain or the C-terminal of the α subunits [39,40,41]. Ardie et al. demonstrated that *KputB1* from *P. tenuiflora* and *KOB1* from rice might regulate K^+^ homeostasis in yeast and *Arabidopsis* [38]. However, the knowledge of voltage-gated K^+^ channel β subunits in abiotic stress tolerance was still limited.

Sweetpotato (*Ipomoea batatas* (L.) Lam) plays an important role in food and bio-energy security [42]. High amounts of K^+^ were needed for yield and quality during the swelling stage of the tuberous root in this crop [43]. Therefore, the uptake and translocation of K^+^ play a vital role in sweetpotato. In a previous study, a total of 11 members of the voltage-gated K^+^ channel family were identified in sweetpotato, among which IbAKT1 (α subunit of K_in_ channels) was demonstrated to regulate the tolerance of K^+^ deficiency [43]. Nevertheless, isolation and functional analyses of voltage-gated K^+^ channel β subunits, which also play important roles in K^+^ transportation in plants, are scarce in sweetpotato.

In this study, a novel gene encoding a voltage-gated K^+^ channel β subunit was identified from the low-K^+^-tolerant sweetpotato variety Shangshu19. The expression level of this gene was significantly higher in Shangshu19 (low-K^+^ tolerant) than that in Yuzi7 (low-K^+^ sensitive) under low-K^+^ conditions. The overexpression of *KIbB1* enhanced tolerance to K^+^ deficiency and high-salinity stresses. This study provides preliminary evidence for the function of *KIbB1* in the tolerance of high salinity and low K^+^. Meanwhile, this information will play an important role in the further analysis of the mechanisms of voltage-gated K^+^ channel β subunits and the functional differences between voltage-gated K^+^ channel α subunits and β subunits in sweetpotato and other plants.

## 2. Materials and Methods

### 2.1. Plant Materials

The expression patterns of *KIbB1* were analyzed in the low-K^+^-tolerant sweetpotato variety Shangshu19 and the low-K^+^-sensitive sweetpotato variety Yuzi7 under both low-K^+^ and normal conditions. The low-K^+^-tolerant sweetpotato variety Shangshu19 was employed for isolating the coding sequence (CDS) of *KIbB1*. *Arabidopsis* (Columbia-0) was used to generate *KIbB1*-overexpression lines. The growth conditions of sweetpotato and *Arabidopsis* were both set according to Zhu et al. [44].

### 2.2. Sequence Analysis of KIbB1

Total RNA was extracted from Shangshu19 using the TsingZol Total RNA Extraction Reagent Kit (Tsingke Biotechnology Co., Beijing, China). Primescript™ RT Reagent Kit with gDNA Eraser (TaKaRa, Beijing, China) was employed to synthesize the first-strand cDNA according to the manufacturer’s protocols. The coding sequence (CDS) of *KIbB1* was amplified with the specific primer KIbB1-F/R (Appendix A). The website ExPASy (https://web.expasy.org/protparam/ accessed on 25 September 2021) was used to predict the molecular weight, isoelectric point and polarity of KIbB1. Protein sequence alignment of KIbB1 and voltage-gated K^+^ channel β subunits from other plant species was conducted with Jalview software (http://www.jalview.org/ accessed on 25 September 2021). The transmembrane domain of KIbB1 was predicted with the online analysis tool TMHM (http://www.cbs.dtu.dk/services/TMHMM-2.0/ accessed on 25 September 2021).

### 2.3. Expression Analysis of KIbB1

*KIbB1* expression levels were analyzed in leaf, stem, fibrous root and tuberous root of 90-day-old field-grown Shangshu19. For induction analysis under low-K^+^ stress, the expression profiles of *KIbB1* were detected in roots and shoots of 4-week-old Shangshu19 and Yuzi7 plantlets grown on Murashige and Skoog (MS) medium. These two sweetpotato varieties were treated with Hoagland solution containing 0 or 20 mM K^+^ for 0, 6, 12, 24 and 48 h. The gene expression analysis was conducted using quantitative real-time polymerase chain reaction (qRT-PCR) with TB Green Premix Ex TaqTM II Kit (TaKaRa, Beijing, China) and QuantStudio 3 (Applied Biosystems, Foster, CA, USA) as described by Zhu et al. [44]. The primer sequences of *IbActin* (internal control) and KIbB1-q-F/R are listed in Appendix A.

### 2.4. Generation of KIbB1 Transgenic Plants

The whole open reading frame (ORF) with restriction enzyme cutting sites of *KIbB1* was amplified with the specific primer KIbB-OE-F/R (Appendix A). After digestion with restriction enzymes, *KIbB1* was inserted into the pCAMBIA1300 vector. The *Agrobacterium tumefaciens* strain GV3101 containing fusion constructs was transferred into *Arabidopsis,* according to Clough et al. [45]. The positive transgenic plants were selected with 50 mg/L hygromycin and identified by PCR using the specific primer KIbB-T-F/R (Appendix A).

### 2.5. Low-K^+^ and High-Salinity Treatment Settings

The *KIbB1* transgenic lines were selected with hygromycin to be T3 homozygous for further analysis. After surface sterilization, seeds from wild-type (WT) and transgenic lines were germinated on 1/2 MS medium. For low-K^+^ treatment, seedlings with 1 cm root length were treated with low K^+^ (50 μM) or normal K^+^ (10 mM) conditions upside down for 10 days. The new downward-curving roots of the tested plantlets were measured [29].

For high-salinity treatment, WT and transgenic lines were transferred to 1/2 MS medium with 0 or 125 mM NaCl after germination on 1/2 MS medium. After two weeks of treatment, the fresh weight and root length of the tested lines were measured. After being grown on 1/2 MS medium for 10 days, seedlings were transferred to potting soil mixture (rich soil: vermiculite = 3:1, *v*/*v*). WT and transgenic plants were irrigated with H_2_O or 300 mM NaCl solution once every 3 days for 2 weeks [46].

### 2.6. Photosynthetic Characteristics Measurement

To measure photosynthetic characteristics, the pot-grown plantlets under different conditions were kept in the dark for 40 min. The IMAG-MAXI (Heinz Walz, Effeltrich, Germany) was employed to detect the maximal photochemical efficiency of photosystem II (PSII) in the dark (Fv/Fm) and the photochemical efficiency of PSII in the light (Fv’/Fm’).

### 2.7. Expression Analysis of Stress-Responsive Gene Quantification

RNA isolation and first-strand cDNA synthesis of transgenic and WT *Arabidopsis* were performed as described above. The expression profiles of stress-responsive genes in the whole plantlets of transgenic and WT lines under normal and high-salinity conditions were analyzed with qRT-PCR assay according to the method of Zhu et al. [44]. The expression level was normalized with the *Arabidopsis* actin gene, and the specific primers are listed in Appendix A.

### 2.8. ROS Accumulation and Scavenging Assay

The leaves of transgenic and WT plants were employed to analyze the accumulation of H_2_O_2_ and O_2_^−^ under different conditions with 3,3′-diaminobenzidine (DAB) and nitro-blue tetrazolium chloride (NBT) staining, respectively [47,48]. Superoxide dismutase (SOD) and peroxidase (POD) activity and proline and MDA content assays were conducted according to the manufacturer’s instructions using the corresponding chemical kit (Suzhou Grace Biotechnology Co, Suzhou, China).

### 2.9. Na^+^ and K^+^ Content Measurement

For K^+^ and Na^+^ concentration measurement, samples were collected and washed with double-distilled water. Samples were dried at 106 °C for 24 h and were dissolved in HNO_3_ and HClO_4_ for 12 h, subsequently. After dissolving on a temperature-controlled heating plate, K^+^ and Na^+^ content was measured using an ICO-OES (OPTMA8000DV; PerkinElmer) according to the manufacturer’s instructions. For shoot/root K^+^ ratio measurement, the shoots and roots of the tested plants were harvested separately.

### 2.10. Statistical Analysis

Each treatment contained three independent biological replicates. All data are presented as the means ± SE. SPSS 17.0 software was employed to perform statistical analysis with a Student’s *t*-test. The graphs were created with GraphPad Prism 8.0 software. * and ** indicate a significant difference at *p* < 0.05 and *p* < 0.01, respectively.

## 3. Results

### 3.1. Sequence Analysis of KIbB1

The ORF of *KIbB1* was isolated from the sweetpotato variety Shangshu19, which was a low-K^+^-tolerant variety (unpublished data). The ORF of this gene was 990 bp, encoding 329 amino acids, with a molecular weight of 36.677 kDa and an isoelectric point (pI) of 8.18. Multiple alignments of KIbB1 and its homologs from *A. thaliana, Oryza sativa, Puccinellia tenuiflora, Egeria densa, Indosa sinica, Hodeum vulgare* and *Triticum dicoccoides* indicated that voltage-gated K^+^ channel β subunits were highly conserved among different plant species (Appendix A). KIbB1 contained seven putative phosphorylation sites and two putative glycosylation sites (Appendix A). KIbB1 polarity analysis showed an overall hydrophilic nature of this protein (Appendix A). Accordingly, no predicted transmembrane domain was found in KIbB (Appendix A).

### 3.2. Expression Profiles of KIbB1

Specific expression analysis of *KIbB1* in different tissues showed that this gene was expressed in all the tested tissues, including the stem, leaf, fibrous root and tuberous root of Shangshu19 (Appendix A). The highest *KIbB1* expression level was found in the fibrous root, which is an important tissue for K^+^ uptake (Appendix A). The EST of *KIbB1* was differently expressed in sweetpotato transcriptome data under salt stress treatment (unpublished data). As a unit of voltage-gated K^+^ channel, the expression of this gene in response to low-K^+^ stress was also analyzed. Under normal conditions, the expression level of *KIbB1* increased with treatment time over 48 h and showed no significant difference between Shangshu19 (low-K^+^ tolerant) and Yuzi7 (low-K^+^ sensitive) (Figure 1a,b). Under low-K^+^ conditions, the expression of *KIbB1* was significantly down-regulated in both Shangshu19 and Yuzi7 (Figure 1a,b). The drift of *KIbB1* expression was similar to that under normal conditions (Figure 1a,b). However, significant differences in *KIbB1* expression were found between Shangshu19 and Yuzi7 under low-K^+^ conditions (Figure 1a,b). These results indicate that *KIbB1* expression might be involved in the K^+^ deficiency response in sweetpotato.

### 3.3. KIbB1 Positively Regulates Low-K^+^ Tolerance in Transgenic Plants

After screening with hygromycin and identification by PCR, a total of 16 homozygous *KIbB1* overexpression *Arabidopsis* lines were obtained for further analysis (Appendix A). WT and *KIbB1* transgenic seedlings were cultured upside-down on 1/2 MS medium with different concentrations of K^+^ (50 μM and 10 mM) for 10 days after germination. Under normal conditions, the new roots of WT and *KIbB1* overexpression lines curved downward (due to geotropism) and showed no obvious morphological difference (Figure 2a). However, the growth of roots was seriously suppressed under low-K^+^ conditions in both WT and transgenic plants (Figure 2a). The rate of root bending and the length of both WT and transgenic lines were significantly lower under K^+^-deficient conditions than under normal conditions (Figure 2b,c). Meanwhile, the length and rate of root bending in transgenic lines were significantly higher than those of WT under low-K^+^ conditions (Figure 2b,c). These results suggest that the overexpression of *KIbB1* enhanced the tolerance of K^+^ deficiency in *Arabidopsis*.

### 3.4. Overexpression of KIbB1 Enhances High-Salinity Resistance in Arabidopsis

Since the expression of *KIbB1* was strongly induced by high-salinity and low-K^+^ stress conditions, the function of this gene in salt stress tolerance was also investigated. On 1/2 MS medium, both WT and transgenic lines grew well and showed no morphological difference (Figure 3a). However, the development of shoot and root was strongly suppressed on 1/2 MS medium with 125 mM NaCl (Figure 3a). The measurement of root length and fresh weight showed no significant difference between WT and transgenic lines under normal conditions (Figure 3b,c). On the other hand, the root length and fresh weight of WT were obviously lower than those of *KIbB1* overexpression lines under high-salinity conditions (Figure 3b,c). Meanwhile, the plantlets grown in pots with soil mixture were also employed for the salt tolerance assay. Under normal conditions, WT and *KIbB1* overexpression lines grew in a similarly exuberant manner (Figure 4a). After saltwater irrigation, wilting leaves were found in both WT and transgenic plants (Figure 4a). However, the growth condition of *KIbB1* overexpression lines was still better than that in WT (Figure 4a). The measurement of photosynthetic characteristics indicated that Fv/Fm was similar among the tested plants under normal conditions (Figure 4b). Fv/Fm mildly decreased in *KIbB1* overexpression lines but strongly decreased in WT under high-salinity conditions (Figure 4b). Fv’/Fm’ in WT and transgenic lines showed a similar drift to Fv/Fm under different treatments (Figure 4c). The photosynthetic characteristics were consistent with the morphology index under normal and salt stress conditions. These results suggest that *KIbB1* positively regulates salt stress resistance in transgenic *Arabidopsis*.

### 3.5. KIbB1 Promotes ROS Homeostasis under High-Salinity Conditions

The stress-induced accumulation of ROS leads to numerous deleterious influences on plants [49]. Therefore, ROS content under different conditions is an important selective index of plants when it comes to enduring high salinity. Several ROS-scavenging genes, such as AtP5CR and *AtP5CS*, belonging to the non-enzymatic system, and *AtSOD*, *AtPOD*, *AtCAT*, *AtAPX*, *AtDHAR* and *AtGPX8*, belonging to the enzymatic system, were analyzed under different treatments. The whole plantlets of pot-grown WT and transgenic lines were employed to analyze the gene expression profiles. The results show that no obvious difference was found in the expression level of the tested genes between WT and *KIbB1* overexpression lines under optimal growth conditions (Figure 5). After salt stress treatment, the expression of these genes was significantly higher in transgenic lines than in WT (Figure 5). These results indicate that the overexpression of *KIbB1* could induce the expression of genes related to ROS scavenging, including enzymatic and non-enzymatic systems, under salt stress treatment.

The activity of SOD and POD, two important enzymes belonging to enzymatic ROS scavenging, was determined under different conditions. Under normal conditions, the activity of SOD and POD showed no significant difference (Figure 6a,b). On the other hand, the activity of SOD and POD was at least 60% and 70.05% higher, respectively, in *KIbB1* overexpression lines than in WT lines under high-salinity conditions (Figure 6a,b). Meanwhile, the content of proline, an element belonging to non-enzymatic ROS scavenging, was also significantly higher in transgenic plants than that in WT (Figure 6c). The content of MDA in transgenic lines was significantly lower than that in WT under salt stress conditions, indicating the degree of membrane damage caused by ROS (Figure 6d). Finally, the accumulation of ROS, including H_2_O_2_ and O_2_^−^ in leaves, was measured with DAB and NBT histochemical staining, respectively. The degree of DAB and NBT staining was slight and was similar between WT and transgenic lines under normal conditions (Figure 6e,f). However, the accumulation of H_2_O_2_ and O^2−^ in WT was obviously more than that in transgenic leaves (Figure 6e,f). These results collectively demonstrate that the overexpression of *KIbB1* enhanced ROS scavenging to positively regulate high-salinity resistance.

### 3.6. KIbB1 Enhances K^+^ Homeostasis under Low-K^+^ and High-Salinity Conditions

As β subunits of voltage-gated K^+^ channels, the function of *KIbB1* in regulating K^+^ uptake and translocation under different conditions was analyzed to reveal the mechanism of this gene in low-K^+^ and high-salinity stresses. Under normal conditions, the concentration of K^+^ and Na^+^, the K^+^/Na^+^ ratio and the shoot/root ratio of K^+^ all showed no significant difference between WT and transgenic lines (Figure 7a–d). The K^+^ concentration in WT and transgenic lines decreased under low-K^+^ and high-salinity conditions, which indicated that the high concentration of Na^+^ affected the K^+^ uptake (Figure 7a). The K^+^ concentration of WT was significantly lower than that of transgenic lines, which suggested that the overexpression of *KIbB1* enhanced the ability of K^+^ acquisition under ion stress (Figure 7a). The variation in Na^+^ concentration was opposite to that of K^+^ concentration. The Na^+^ concentration in WT and transgenic lines increased, and it was higher in WT than that in transgenic lines under ion stress conditions (Figure 7b). The results indicate that the absorption of Na^+^ was promoted by K^+^ deficiency and could be suppressed by the overexpression of *KIbB1*. Accordingly, the K^+^/Na^+^ ratio in transgenic plantlets was significantly higher than that in WT (Figure 7c). The shoot/root ratio of K^+^ was much higher in overexpression lines than that in WT under low-K^+^ stress; however, there was no significant difference among the tested lines under high-salinity conditions (Figure 7d). The results demonstrate that *KIbB1* functioned in K^+^ translocation from root to shoot under K^+^-deficient conditions but not under salt stress. These results collectively demonstrate that the overexpression of *KIbB1* helped to maintain K^+^ homeostasis under low-K^+^ and high-salinity conditions in plants, but the mechanisms were a little different.

## 4. Discussion

As one of the most important nutrients, K^+^ participates in many fundamental processes, such as enzyme activation, anion homeostasis and osmotic adjustment [50]. Plant root cells need to uptake K^+^ from the environment against the concentration gradient since the K^+^ concentration in root cells is much higher than that in soil [16]. The acquisition and translocation of K^+^ are regulated by K^+^ transporters and channels. The voltage-gated K^+^ channel is an important K^+^ channel known to be found in both animals and plants, mediating K^+^ flux [19]. The structure of voltage-gated K^+^ channels in plants is similar to that in animals. Several studies demonstrated that voltage-gated K^+^ channels are homotetramers consisting of four pore-forming α subunits, which is sufficient to induce K^+^ currents across membranes [37]. Subsequently, numerous findings showed the complexity of these proteins. Since the β subunits of voltage-gated K^+^ channels were identified in the bovine brain, it has been demonstrated that some of the voltage-gated K^+^ channels have a second structural component in many species [35]. The first voltage-gated K^+^ channel β subunits in plants were isolated from *Arabidopsis,* and several additional voltage-gated K^+^ channel β subunits have been identified in different plants since [37,38,51]. However, few studies have been conducted on voltage-gated K^+^ channel β subunits in sweetpotato until now. In this study, we isolated a gene encoding a specific protein in sweetpotato, which shared high conservation with KAB1 from *Arabidopsis* and KOB1 from rice (Appendix A). Therefore, we speculated to be a voltage-gated K^+^ channel β subunit named KIbB1.

Owing to the limited availability of K^+^ in soil, K^+^ deficiency is a common source of abiotic stress for plants [12]. Plants cope with K^+^ deficiency through sensing changes in K^+^ concentration and transmitting signals to modulate their physiology and morphology [52]. The plant root system is an important organ for K^+^ absorption; therefore, the root morphology is influenced by the level of K^+^ content [53]. Several reports have suggested that the growth of roots is repressed under low-K^+^ conditions and that the K^+^-efficient variety is less inhibited than the K^+^-sensitive variety [54,55,56]. Previous studies demonstrated that factors associated with K^+^ acquisition and translocation could enhance low-K^+^ tolerance by maintaining cation homeostasis [57]. The expression of *AtAKT1* and *AtHAK5* was strongly induced by low-K^+^ conditions, and these two genes played important roles in low-K^+^ tolerance by contributing to K^+^ acquisition [57]. *GmAKT1* positively regulated K^+^ uptake to enhance the development of roots under K^+^-starvation treatment [32]. The overexpression of *IbAKT1* showed higher K^+^ influx and lower K^+^ efflux than WT under low-K^+^ conditions [43]. KputB1, a voltage-gated K^+^ channel β subunit in *P. tenuiflora*, increased K^+^ content in the shoot in order to enhance ionic balance under K^+^ deficiency in *Arabidopsis* [38]. In our study, the expression of *KIbB1* showed no significant difference under normal conditions between Shangshu19 and Yuzi7; however, it was evidently higher in Shangshu19 than that in Yuzi7 under low-K^+^ conditions (Figure 1a,b). Therefore, we speculate that *KIbB1* plays a crucial role in low-K^+^ stress resistance. To verify this assumption, *KIbB1* transgenic plants were employed for further functional identification. The overexpression of *KIbB1* significantly enhanced the rate and length of roots bending under K^+^-starvation stress (Figure 2a–c). Further ion content measurement showed that the concentration of K^+^, the K^+^/Na^+^ ratio and the shoot/root ratio of K^+^ were significantly higher, and the concentration of Na^+^ was significantly lower in transgenic lines than those in WT under low-K^+^ conditions (Figure 7a–d). These results agree with those of previous studies and indicate that *KIbB1* positively regulates low-K^+^ tolerance through mediating K^+^ uptake and translocation.

High salinity is a major abiotic stress, and the adverse effects on crop production have been demonstrated in numerous studies [58,59,60]. High salinity induces several negative impacts on physiological and biochemical mechanisms such as osmotic stress, ionic imbalance, photosynthetic destructiveness and oxidation damage [7,61]. The accumulation of Na^+^ under salt stress inhibits K^+^ uptake, affecting the K^+^/Na^+^ ratio [62]. Meanwhile, the excessive generation of ROS is a common result of salt stress in plants, which may induce cellular damage and death [63]. Plants have evolved a range of mechanisms to cope with high-salinity stress and ensure their survival [64]. Several previous studies showed that genes involved in ion homeostasis, osmolyte synthesis and ROS scavenging played important roles in salt stress tolerance [61]. K^+^ channels are widely reported to participate in the acquisition and translocation of K^+^, for which their functions in salt stress tolerance have also been studied. In barley, *HvAKT1* helps to enhance the Na^+^/K^+^ balance of mesophyll cells responding to high-salinity conditions [31]. *GmHKT1;4* could regulate the Na^+^/K^+^ ratio in roots to enhance salt stress in soybean [65]. Wu et al. [66] demonstrated that KOR (K^+^ outward rectifying) channels and NSCC (non-selective cation) channels mediating NaCl-induced K^+^ efflux in leaf mesophyll could be increased in the presence of ROS. For a voltage-gated K^+^ channel β subunit, Ardie et al. [38] found that *KputB1* increased K^+^ content under high-salinity stress. In the present study, the overexpression of *KIbB1* improved morphology conditions and photosynthetic characteristics, including Fv/Fm and Fv’/Fm’, under salt stress (Figure 3 and Figure 4). This suggests that *KIbB1* positively regulated salt stress tolerance in transgenic *Arabidopsis*. The expression levels of genes related to ROS scavenging were much higher than those in WT after salt stress treatment (Figure 5). The content of proline and the activity of SOD and POD were also significantly higher in transgenic lines (Figure 6a–c). Accordingly, the accumulation of ROS such as H_2_O_2_ and O_2_^−^ was lower in transgenic leaves than in WT (Figure 6e,f). It has been demonstrated that genes related to antioxidant enzymes could enhance the tolerance of salinity stress by scavenging the excessive accumulation of ROS induced by salt stress [67]. These results indicate that the overexpression of *KIbB1* activated ROS scavenging under salt stress. Further ion content measurement showed that the variation in K^+^ and Na^+^ concentration and K^+^/Na^+^ ratio under high-salinity stress was similar to that under low-K^+^ conditions; however, the shoot/root ratio of K^+^ showed no significant difference among the tested lines (Figure 7a–d). These results collectively demonstrate that *KIbB1* may be involved in the salt stress response by enhancing K^+^ uptake to mediate Na^+^/K^+^ homeostasis. This study provides gene resources for improving high-salinity and low-K^+^ tolerance by genetic engineering in sweetpotato. There are two directions for future research: the different molecular mechanisms of *KIbB1* under high-salinity or low-K^+^ conditions; and the functional differences between voltage-gated K^+^ channel α subunits and β subunits in K^+^ transportation.

## 5. Conclusions

In conclusion, we isolated a novel gene encoding a voltage-gated K^+^ channel β subunit from sweetpotato, named *KIbB1*. The expression of this gene was significantly higher in the low-K^+^-tolerant line than that in the low-K^+^-sensitive line under K^+^-starvation stress. The overexpression of *KIbB1* enhanced resistance to K^+^ deficiency through decreasing Na^+^ absorption and improving K^+^ uptake and translocation in *Arabidopsis*. Meanwhile, *KIbB1* also positively regulated high-salinity tolerance. ROS scavenging was activated to decrease the salt-induced ROS accumulation. The measurement of ion concentration showed that *KIbB1* regulated the acquisition of Na^+^ and K^+^ but not the translocation. These results indicate that *KIbB1* enhanced low-K^+^ and high-salinity tolerance through maintaining the K^+^/Na^+^ balance in plants, although the mechanisms are a little different.

## Figures and Tables

**Figure 1 genes-13-01100-f001:**
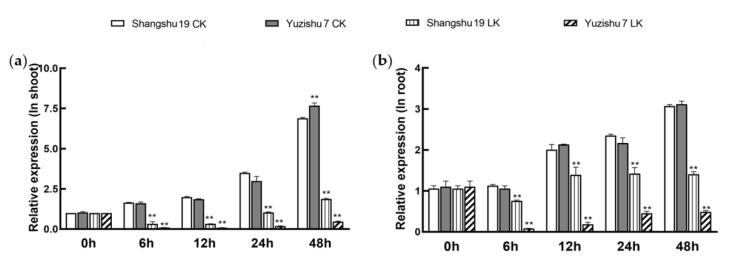
Expression analysis of KIbB1 under normal and low-K^+^ conditions in (**a**) shoots and (**b**) roots. CK means normal condition, and LK means low-K^+^ treatment. Data are presented as means ± SE (*n* = 3). ** indicates a significant difference at *p* < 0.01.

**Figure 2 genes-13-01100-f002:**
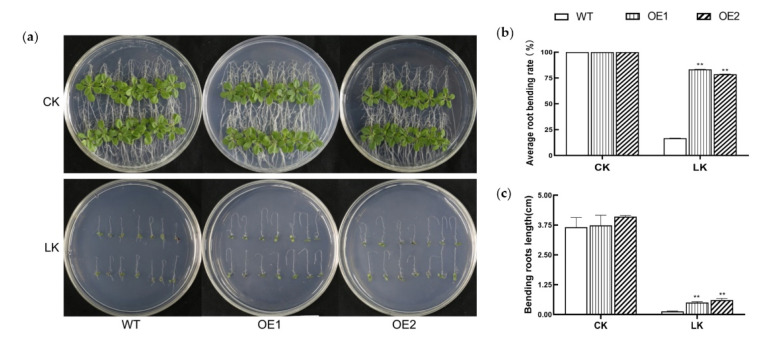
Responses of KIbB1 transgenic and WT seedlings cultured on 1/2 MS medium with normal and low-K^+^ conditions. (**a**) Phenotypes of the tested plants. Rate (**b**) and length (**c**) of root bending of the tested plants under different conditions. Data are presented as means ± SE (*n* = 3). ** indicates a significant difference at *p* < 0.01.

**Figure 3 genes-13-01100-f003:**
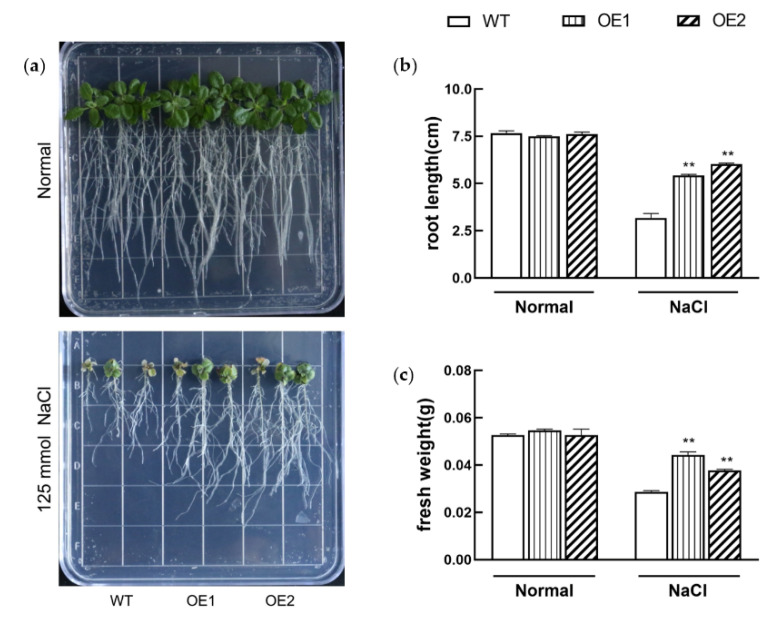
Responses of transgenic and WT seedlings cultured on normal 1/2 MS medium or under high-salinity conditions. (**a**) Morphology analysis of transgenic and WT plants. (**b**) Primary root length. (**c**) Fresh weight. Data are presented as mean ± SE (*n* = 3). ** indicates a significant difference at *p* < 0.01.

**Figure 4 genes-13-01100-f004:**
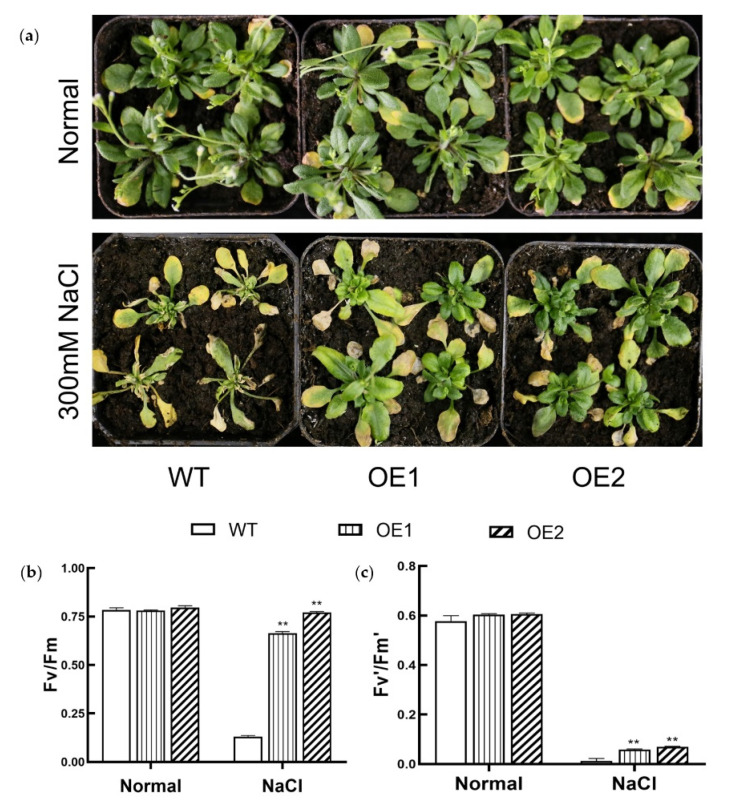
Responses of transgenic and WT plants grown in pots under normal and high-salinity conditions. (**a**) Morphological analysis of tested plants under different conditions. Photosynthetic characteristics of tested plants include (**b**) the maximal photochemical efficiency of photosystem II (PSII) in the dark (Fv/Fm) and (**c**) the photochemical efficiency of PSII in the light (Fv’/Fm’). Data are presented as mean ± SE (*n* = 3). ** indicates a significant difference at *p* < 0.01.

**Figure 5 genes-13-01100-f005:**
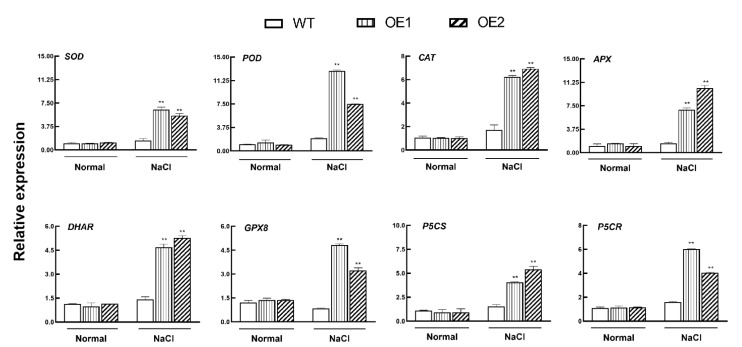
Expression analysis of genes involving in ROS scavenging under normal and high-salinity conditions in transgenic and WT plants. Data are presented as mean ± SE (*n* = 3). ** indicates a significant difference at *p* < 0.01.

**Figure 6 genes-13-01100-f006:**
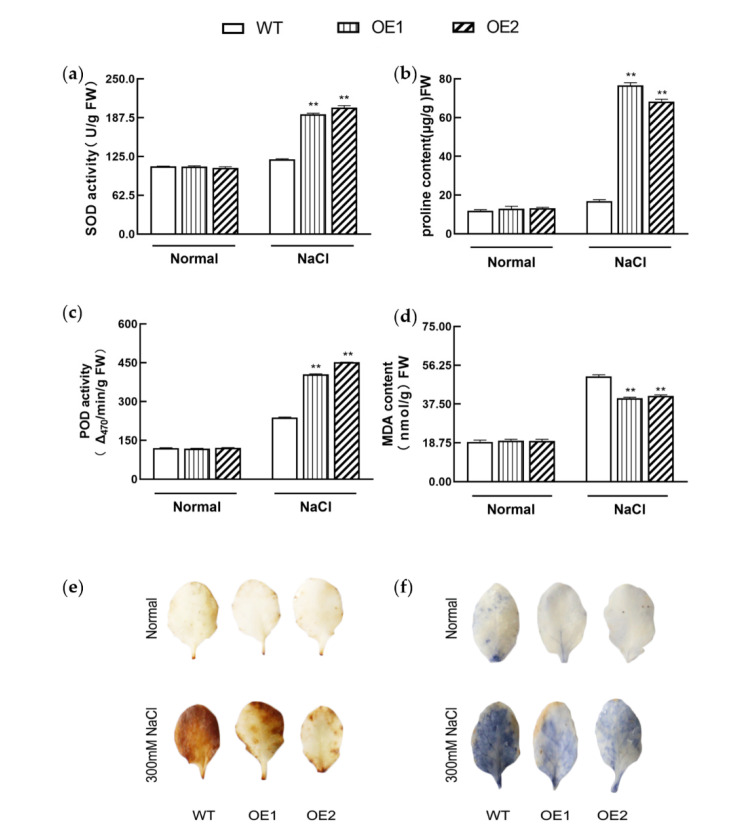
ROS scavenging and accumulation analysis under normal and high-salinity conditions in the tested lines. (**a**) SOD activity. (**b**) POD activity. (**c**) Proline content. (**d**) MDA content. Data are presented as means± SD (*n* = 3). ** indicates a significant difference at *p* < 0.01. (**e**) DAB and (**f**) NBT staining of transgenic and WT leaves.

**Figure 7 genes-13-01100-f007:**
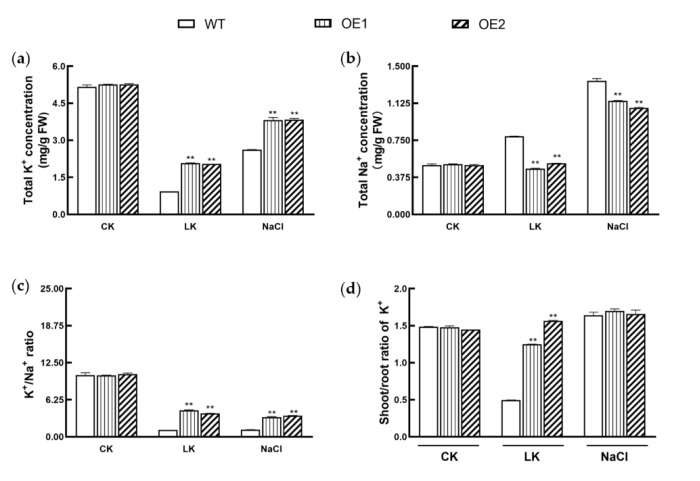
Measurement of K^+^ and Na^+^ in transgenic and WT plants under normal, low-K^+^ and high-salinity conditions. (**a**) K^+^ and (**b**) Na^+^ concentration of the whole plantlets. (**c**) K^+^/Na^+^ ratio of the whole plants. (**d**) Shoot/root ratio of K^+^. Data are presented as means ± SE (*n* = 3). ** indicates a significant difference at *p* < 0.01 between WT and transgenic lines.

## Data Availability

Not applicable.

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
