# Peer review of "The Sweetpotato Voltage-Gated K+ Channel β Subunit, KIbB1, Positively Regulates Low-K+ and High-Salinity Tolerance by Maintaining Ion Homeostasis"

_genes, 2022, doi:10.3390/genes13061100_

Round 1

Reviewer 1 Report

Dear Authors

The present manuscript entitled “The sweetpotato voltage-gated K+ channel β subunit, KIbB1, positively regulates low K+ and high salinity tolerance by maintaining ion homeostasis” demonstrated physiological function of a novel gene KIbB1 encoding a voltage-gated K+ channel β subunit in sweet potato and suggested that the mechanisms of KIbB1 in regulating K+ were somewhat different between low K+ and high salinity conditions. The investigation has been planned very well and results are quite appealing, although there are some opportunities for further improvement, please find the suggestions below.

1.       Abstract- line 25- the KIbB1 studies seems to be performed in transgenic line, please provide some information regarding this.

2.       Line 40- Numerous geographical mechanism, please be specific regarding it.

3.       Line 58-89, This text suits better for discussion.

4.       Line 112- Please include more details regarding KIbB1-overexpression lines.

5.       Line 167- please include the conditions for qRT-PCR.

6.       Legend for figure 1 should be self-explanatory, please include what is no 19, no 7, CK, LK?

7.       Line 418- please explain that KIbB1 from sweetpotato is a novel gene?

Thank you

Author Response

  1. Abstract line 25- the KIbB1 studies seems to be performed in transgenic line, please provide some information regarding this.

Response: The function analysis of KIbB1 was conducted in transgenic Arabidopsis planelets. The generation of KIbB1 overexpression lines was described in the Materials and Methods Section 2.4.

  1. Line 40- Numerous geographical mechanism, please be specific regarding it.

Response: We mean that the environmental factors such as salinity stress can affect the geographical distribution of plants. The previous statement was not appropriate and we had revised the sentence. (Introduction Section, Page 1, Line 37-39)

  1. Line 58-89, This text suits better for discussion.

Response: Thanks for your suggestions. Since the knowledge of β subunits of the voltage-gated K+ channels is limited, we provided some related information of this kind of genes in plants in the Introduction Section.

  1. Line 112- Please include more details regarding KIbB1-overexpression lines.

Response: The KIbB1 transgenic Arabidopsis used in this research were generated using the method of Clough et al (1998). The primers, vectors and strains used in this study were described in the Materials and Methods Section 2.4. All the transgenic lines have been identified by PCR with specific primers (Table S1, Figure S3).

  1. Line 167- please include the conditions for qRT-PCR.

Response: The conditions for qRT-PCR were set as Zhu et al. (2020) described. We have added this information in the revised manuscript. (Materials and Methods Section, Page 4, Line 169)

  1. Legend for figure 1 should be self-explanatory, please include what is no 19, no 7, CK, LK?

Response: Thanks for your suggestion. We have revised the legend for Figure 1. Shangshu19 and Yuzi7 are two sweetpotato varieties. CK means normal K+ condition and LK means low K+ condition. (Results Section, Page 5, Line 221-222; Page 5, Figure 1)

  1. Line 418- please explain that KIbB1 from sweetpotato is a novel gene?

Response: We combed through the reports in sweetpotato and found no related research of KIbB1, so we thought that the KIbB1 was a novel gene regulating salt and low K+ stress tolerance from sweetpotato.

Reviewer 2 Report

The manuscript “The sweet potato voltage-gated K+ channel β subunit, KIbB1, positively regulates low K+ and high salinity tolerance by maintaining ion homeostasis” deals with the study of various K transported, expression of transported/ channel, and antioxidant activities of the transgenic sweet potato gene in Arabidopsis. Overall the manuscript is well written, planned, and executed. I have enjoyed reading this research manuscript, and indeed this manuscript will help the researchers to develop salt-tolerant cultivars. However, some minor issues need to be addressed before final decision is made.

Comments:

1.     LN 39-40: Please rewrite with latest reference

https://doi.org/10.3390/life11060545

https://doi.org/10.1007/s00425-022-03845-y

2.     LN 87: Please make the scientific word italics.

3.     LN 90: Please make the scientific word italics.

4.     LN 92-93: This line do not convey any meaning in this paragraph. Kindly remove it.

5.     The objective of the research is not properly defined. Kindly rewrite the objective of the research work in the last paragraph of the introduction section.

6.     Section 2.10: Please also mention the software name in which the graphs were made.

7.     Was there any prior screening of taking salinity level as 300 mM NaCl? Or it is based on some literature? Please explain.

8.     Fig. 5: There are no legends in y axis.

9.     In section 3.5, the author should also mention the percentage increase or decrease in the antioxidant enzyme activity.

10.  In discussion section  LN 404: The author should also discuss on the interaction of ROS and antioxidant enzymes

https://doi.org/10.1007/s00344-022-10591-8

11.  In the discussion section, the author should also discuss about the SOS pathway in relation to their experiment.

12.  In the conclusion section, the authors should also write a few lines on the future impact of this study and on which further line the researcher needs to focus. 

Author Response

LN 39-40: Please rewrite with latest reference

https://doi.org/10.3390/life11060545

https://doi.org/10.1007/s00425-022-03845-y

Response: We have rewritten this sentence and cited the latest reference. (Introduction Section, Page 1, Line 39-40)

  1. LN 87: Please make the scientific word italics.

Response: Thanks for your suggestion. We have revised this mistake and have checked the scientific word throughout the manuscript. (Introduction Section, Page 2, Line 78)

  1. LN 90: Please make the scientific word italics.

Response: We have revised this mistake and have checked the scientific word throughout the manuscript. (Introduction Section, Page 2, Line 89)

  1. LN 92-93: This line does not convey any meaning in this paragraph. Kindly remove it.

Response: We have removed this sentence.

  1. The objective of the research is not properly defined. Kindly rewrite the objective of the research work in the last paragraph of the introduction section.

Response: Thanks for your suggestion. We have rewritten the objective of this study in the revised manuscript. (Introduction Section, Page 2, Line 92-96; Page 3, Line 102-103)

  1. Section 2.10: Please also mention the software name in which the graphs were made.

Response: We made the graphs using the software named GraphPad Prism 8 and we have added this information in the revised manuscript. (Materials and Methods Section, Page 4, Line 190)

  1. Was there any prior screening of taking salinity level as 300 mM NaCl? Or it is based on some literature? Please explain.

Response: We chose salinity level as 300 mM NaCl according to our several previous studies. This level of salt treatment could inhibit the growth of wide type Arabidopsis.

  1. Fig. 5: There are no legends in y axis.

Response: We have revised this figure. (Results Section, Page 8, Figure 5)

  1. In section 3.5, the author should also mention the percentage increase or decrease in the antioxidant enzyme activity.

Response: Thanks for your suggestion. This modification could show the changes of the antioxidant enzyme activity clearly. We have added the percentage increase in the antioxidant enzyme activity in the revised manuscript. (Results Section, Page 8, Line 300-301)

  1. In discussion section LN 404: The author should also discuss on the interaction of ROS and antioxidant enzymes.

https://doi.org/10.1007/s00344-022-10591-8

Response: Thanks for your suggestion. We have discussed the interaction of ROS and antioxidant enzymes in the revised manuscript and cited the latest reference. (Discussion Section, Page 11, Line 421-423)

  1. In the discussion section, the author should also discuss about the SOS pathway in relation to their experiment.

Response: Thanks for your good suggestion. SOS pathway plays important roles in ion homeostasis maintaining under salt stress. In our present study, we have found little evidence that the overexpression of this gene strongly effects SOS pathway. We will conduct more research in this area in the future.

  1. In the conclusion section, the authors should also write a few lines on the future impact of this study and on which further line the researcher needs to focus.

Response: Thanks for your suggestion. We have added sentences related to future impact of this study. But we added this information in the Discussion Section. (Discussion Section, Page 12, Line 429-434)